# The Interaction between Hydromulching and Arbuscular Mycorrhiza Improves Escarole Growth and Productivity by Regulating Nutrient Uptake and Hormonal Balance

**DOI:** 10.3390/plants11202795

**Published:** 2022-10-21

**Authors:** Miriam Romero-Muñoz, Amparo Gálvez, Purificación A. Martínez-Melgarejo, María Carmen Piñero, Francisco M. del Amor, Alfonso Albacete, Josefa López-Marín

**Affiliations:** 1Institute for Agro-Environmental Research and Development of Murcia (IMIDA), Department of Plant Production and Agrotechnology, C/Mayor s/n, E-30150 Murcia, Spain; 2Centro de Edafología y Biología Aplicada del Segura (CEBAS-CSIC), Department of Plant Nutrition, Campus Universitario de Espinardo, E-30100 Murcia, Spain

**Keywords:** *Rhizophagus irregularis*, arbuscular mycorrhizal fungi, cytokinins, gibberellins, escarole, nutrient use efficiency

## Abstract

To improve water and nutrient use efficiencies some strategies have been proposed, such as the use of mulching techniques or arbuscular mycorrhizal fungi (AMF) inoculation. To gain insights into the interaction between the use of hydromulch and AMF inoculation on plant growth and productivity, escarole plants (*Cichorium endivia*, L.) were inoculated with the AMF *Rhizophagus irregularis* and grown with non-inoculated plants under different soil cover treatments: ecological hydromulching based on the substrate of mushroom cultivation (MS), low-density black polyethylene (PE), and non-covered soil (BS). AMF inoculation or the use of mulching alone, but especially their interaction, increased the plant growth. The growth improvement observed in AMF-inoculated escarole plants grown under hydromulching conditions was mainly associated with the upgrading of nitrogen and phosphorous use efficiency through the regulation of the hormonal balance. Both hydromulching and AMF inoculation were found to increase the active gibberellins (GAs) and cytokinins (CKs), resulting in a positive correlation between these hormones and the growth-related parameters. In contrast, the ethylene precursor 1-aminocyclopropane-1-carboxylic acid (ACC) and abscisic acid (ABA) decreased in AMF-inoculated plants and especially in those grown with the MS treatment. This study demonstrates that there exists a positive interaction between AMF and hydromulching which enhances the growth of escarole plants by improving nutrient use efficiency and hormonal balance.

## 1. Introduction

Escarole (*Cichorium endivia* L.) is one of the most used vegetables in the preparation of ready-to-eat salads, with increasing interest worldwide for its characteristic crunchy texture and mildly bitter taste [1]. In the Mediterranean area, escarole can be grown throughout the year in all crop systems [2], and it is considered a very important crop whose production has increased during the last 10 years, reaching a total world production of 27.66 thousand tons in 2020 [3]. In recent years, the concern about environmental sustainability has increased along with the growing demand for feed and food resources [4]. In this context, current agricultural production practices have to embrace modern agricultural technologies in order to achieve food security for an increasing population. Therefore, much attention has been paid to the use of different agronomic management strategies such as the application of arbuscular mycorrhizal fungi (AMF), which are known to be involved in the improvement of plant growth and crop productivity [5,6,7]. Several studies have shown a direct implication of mycorrhiza in terms of nutrient absorption and translocation due to the extra-radical mycelium that can effectively improve nutrient uptake, thereby improving plant growth and development [8,9]. Indeed, Balliu et al. [10] stated an increase in the leaf area, nitrogen, potassium, calcium, and phosphorous contents which enhanced the plant growth rate in AMF-inoculated tomato plants. Other authors have shown the positive effect of AMF inoculation through the improvement of mineral uptake, chlorophyll synthesis, and water use efficiency under saline conditions [11]. The growth boost due to the mycorrhizal association can be explained by several mechanisms used by fungi under certain conditions [12]. These include the increment of mineral uptake, mainly phosphorous, as well as the production of secondary metabolites such as amino acids, vitamins, and hormones [9,11,13,14]. Concerning phytohormones, their key role has been demonstrated in the regulation of mycorrhizal symbioses [15,16,17]. Various studies have shown the implication of gibberellins (GAs) in the arbuscule formation in plant roots [18], while other early reports suggested that AMF are involved in the upregulation of GAs in tomato plants [19]. Cytokinins (CKs) have been considered an important hormone class related to growth enhancement in horticultural crops [20,21]. In this regard, it has been reported that a stronger increase in the shoot CK content was accompanied by elevated root CK levels associated with a positive plant growth response to AM symbiosis [22,23]. Cosme et al. [24] and Martínez-Medina et al. [25] have found that both shoot- and root-specific alterations of CK levels may be related to growth responses in inoculated tobacco and melon plants.

AMF have also been demonstrated to produce changes in the levels of other hormone classes. Some studies have reported significant changes in the concentrations of the ethylene precursor, 1-aminocyclopropane-1-carboxylic acid (ACC), and indoleacetic acid (IAA) in mycorrhizal plants [25,26], depending on the AM fungus involved. Other studies informed that jasmonic acid (JA) content increased in the roots of mycorrhizal *Medicago truncatula* and soybean seedlings [27,28]. Regarding abscisic acid (ABA), the relation between ABA and the growth of AMF plants has been unclear so far. Some authors have stated that mycorrhization increased ABA concentrations in the host plants [29], while other authors have reported no effects of AMF inoculation on the ABA content [25]. To date, a large number of studies have been conducted to investigate the effects of AMF on plant growth and productivity. Moreover, traditional plastic mulching has also been demonstrated to be beneficial for plant productivity, increasing crop yield and quality [30,31,32]. Nevertheless, there is an urgent need to investigate new alternatives to the use of plastic in crops due to the environmental problems generated by its use [33]. Recently, several authors have reported the benefits of the use of hydromulching, a new semiliquid soil cover formulation, which is based on different biological additives from crop residues [34,35]. Furthermore, the positive effect of the use of different types of hydromulches on plant yield in artichoke plants has been demonstrated [36]. Very recently, Romero-Muñoz et al. [20] have shown a direct implication of the hormonal balance regulation on the growth enhancement of hydromulched escarole plants under drought conditions.

To the best of our knowledge, no studies have been reported so far on the interaction of AMF with sustainable soil cover materials. Considering that both AMF inoculation and hydromulching have been reported to improve plant growth, we hypothesized that their interaction could have a synergic effect on plant productivity explained by the regulation of nutrient use efficiency and hormonal balance.

## 2. Results

### 2.1. Mycorrhizal Colonization and Growth-Related Determinations

The AMF *Rhizophagus irregularis* successfully colonized the roots of the escarole plants evaluated (by 60% on average, Figure 1 and Figure 2A). No AMF colonization was detected in non-mycorrhizal (NM) plants (Figure 1). The use of mulching did not produce any difference in the percentage of root colonization compared to plants without cover.

Plant growth was significantly affected by the application of mulching and AMF. The use of mulching increased total FW, especially in AM plants, but this increment was only significant in escarole plants mulched with MS treatment (by 41%, Figure 2B). Similarly, shoot FW was significantly higher in mycorrhized plants grown under MS treatment in comparison with uncovered plants (by 32%, Figure 2C). Root FW and leaf area significantly increased (by 64% and 29%, respectively) in mycorrhized plants grown with MS treatment (Figure 2D,E). In contrast, although the use of AMF produced a slight increase in the number of leaves in mulched plants, it was not significant (Figure 2F).

### 2.2. Gas Exchange Measurements

The combined use of AMF and mulching also produced differential leaf gas exchange responses (Figure 3). In non-mycorrhizal plants, A_CO_2__ increased over time both in mulched plants and non-mulched plants, but, in general, plants covered with PE presented higher levels throughout the whole period (Figure 3A). Interestingly, the A_CO_2__ also increased over time by the use of AMF, with higher absolute values than those of NM plants, but this increase was only significant in plants mulched with PE at the last time-point. Interestingly, non-mycorrhizal plants presented the lowest values of photosynthesis during the studied period, which were significantly different in non-mulched plants 52 days after transplanting. Stomatal conductance also increased over time, and this increase was especially apparent in AM plants grown under PE treatment at the last analytical time-point (Figure 3B). During the first 30 days of the experimental period, the AMF did not produce any effect in terms of stomatal conductance, though the increase occurred from day 45 onwards. Importantly, although no differences were observed among mulching treatments, mycorrhized plants grown under PE and MS treatments showed higher stomatal conductance at the end of the experimental period (Figure 3B). The transpiration rate was strongly affected by the use of AMF over time (Figure 3C). Notably, this gas exchange parameter reached the maximum value in mycorrhized plants grown under PE treatment, whereas non-mulched plants presented the lowest values of transpiration at the last time point of the considered period (Figure 3C). Intrinsic water use efficiency (WUEi), calculated as the ratio between photosynthesis and transpiration rate, was higher at the beginning of the experiment and decreased during the considered period (Figure 3D). Importantly, mycorrhized plants presented in general the highest levels of WUEi, especially at the end of the experiment, but only non-covered plants presented significant differences compared to non-mycorrhized plants (Figure 3D).

### 2.3. Chlorophyll Content and Fluorescence

Mycorrhized plants mulched with PE presented a slightly higher content of chlorophyll in comparison with the other treatments (by 15% on average), although this increase was not significant (Figure 4A). The same trend was observed in chlorophyll b concentrations (Figure 4B). However, even though no differences were observed in chlorophyll b, an increase was subtly observed in mycorrhized plants (by 8% on average) compared to non-mycorrhized plants. Regarding, chlorophyll fluorescence (Fv/Fm), the use of mulching and AMF inoculation did not produce any effect on this photosynthetic parameter (Figure 4C).

### 2.4. Plant Mineral Content and Nutrient Use Efficiency

The two experimental factors of this study, mulching and AMF inoculation, produced a differential ionic profile in the leaves and roots of escarole plants. Regarding leaves, P^5+^ concentrations were significantly lower in the mycorrhized plants (over 40% on average, Table 1); this can be explained by the use of the Hoagland-P^low^ solution. Interestingly, AMF inoculation increased SO_4_^2−^ and reduced Na^+^ concentrations, but these changes were only significant in PE-mulched plants (Table 1). Furthermore, the use of the AMF or mulching treatment alone as well as their interaction decreased NO_3_^−^, Mg^2+^, and Cu^2+^ concentrations in the leaves of escarole plants (Table 1). Likewise, the inoculation with AMF was found to decrease Na^+^ compared to non-mycorrhized plants.

Importantly, AMF inoculation provoked a general ionic accumulation in the roots of escarole plants (Table 2). Overall, mycorrhized plants showed an increase in K^+^, Mg^2+^, Ca^2+^, Cu^2+^, Mn^2+^, and B^3+^ concentrations, but this increase was only significant in plants mulched with MS (Table 2).

Table 3 shows the nutrient use efficiency of escarole plants. Interestingly, there existed a clear interaction between AMF inoculum and the mulching treatment in nitrogen use efficiency (NUE) and phosphorous use efficiency (PUE). Plants mulched with MS presented a significant increase in NUE both in non-mycorrhizal and mycorrhizal plants in comparison with uncovered plants (by 3.2-fold and 5.6-fold, respectively). Furthermore, the same trend occurred in PUE, but only non-mycorrhizal plants presented significant differences among the mulching treatments (Table 3). Potassium use efficiency (KUE) was not affected by either the mulching treatment or the inoculation with AMF.

### 2.5. Hormonal Profile

Figure 5 shows the GA profile in escarole plants, composed of GA3 and GA4. The balance of the two most active GAs was affected by the use of AMF and the mulching treatment (Figure 5). GA3 was the most abundant GA in escarole leaves. MS treatment provoked a significant increase in the endogenous GA3, but, in this case, it was especially apparent in the AMF-inoculated plants in comparison with inoculated plants without cover (5.7-fold, Figure 5A). GA4 followed the same trend since the use of MS mulch provoked a strong increase in this hormone in mycorrhizal plants in comparison to non-covered plants (7.7-fold, Figure 5B). Consequently, total GA concentrations, calculated as the sum of GA3 and GA4, were significantly higher in mycorrhized plants mulched with MS, while the leaves of non-mulched plants presented the lowest total GA concentrations (Figure 5C).

Three of the most active CKs in higher plants, tZ, RZ, and iP, were analyzed in escarole leaves, but only tZ and iP concentrations were detected (Figure 6). The concentrations of tZ were strongly affected by the use of mulching, both in mycorrhized and non-mycorrhized plants (Figure 6A). The use of AMF provoked an increase both in PE and MS mulched plants (2.0 and 2.2-fold, respectively) in comparison with those without cover. For iP, the highest concentration was also found in the interaction between AMF and MS mulch (by 72%, Figure 6B). Total CK concentrations, calculated as the sum of tZ and iP, were significantly higher in mycorrhized plants mulched with MS treatment (2.3-fold higher than non-covered plants), followed by plants covered with plastic (Figure 6C). Regarding auxins, the most active one in higher plants, IAA, was detected in all treatments. Even though no significant differences were observed between the different mulching treatments, plants mulched with MS presented a significant increase in IAA in the presence of the AMF inoculum (Figure 7A).

ABA has been linked traditionally with a wide range of plant stress responses. ABA concentrations were heterogeneous depending on the mulching treatment and the presence of AMF inoculum (Figure 7B). In non-mycorrhized plants, the lowest concentrations of this hormone were found in PE-mulched plants (30% lower on average). When plants were grown with AMF inoculum, plants without cover had significantly increased ABA concentrations compared to plants mulched with PE (Figure 7B). The important role of ethylene as a stress-related hormone is widely recognized, and its direct precursor, ACC, was measured in the leaves of escarole plants (Figure 7C). The use of AMF decreased ACC concentrations in all mulching treatments, but plants mulched with MS presented significantly lower ACC concentrations (by 30%) than non-mulched plants with or without AMF inoculum. SA concentrations were strongly affected by the use of mulching (Figure 7D) both in mycorrhized and non-mycorrhized plants. Even though no significant differences were observed, plants without cover showed a decrease in SA concentrations both in the presence and absence of AM inoculum in comparison with those mulched with PE and MS (Figure 7D). AMF inoculation also provoked a general increase in JA concentrations, which was more evident in BS and MS plants (Figure 7E). Similarly to SA, the use of mulching affected JA concentrations, especially in mycorrhized plants grown under MS treatment (2.3-fold higher) compared to those without cover (Figure 7E).

To summarize the changes in the hormonal balance of escarole leaves provoked by the combination of different mulching treatments and the inoculation with AMF, a cluster heat plot was performed (Appendix A). As previously mentioned, the hormonal profile studied was clearly affected both by the type of mulch and by the presence or absence of AMF inoculum. The results indicate that GAs and CKs were the hormones that augmented in mycorrhizal plants subjected to mulching treatments, especially in MS plants. In contrast, ACC and ABA decreased in mulched plants, particularly in those inoculated with AMF (Appendix A).

### 2.6. Principal Component Analysis

In order to identify important parameters associated with the variability factors studied, the data set was subjected to a score principal component analysis (PCA, Figure 8A). This statistical test transforms the normalized data into principal component scores through multiple-dimension rotation [37]. The score PCA showed a clear separation between the scores of the escarole plants evaluated with and without AMF inoculum and among the different mulching treatments. The scores of plants grown without AMF inoculum were clearly separated from those with AMF inoculum (Figure 8A). Importantly, MS treatment provoked a strong separation of the scores of mycorrhized plants compared to the other clusters.

Furthermore, a loading PCA was performed to reduce the dimensionality of the data set with the loadings of the variables used in this study (Figure 8B). This mathematical algorithm allows the identification of important ionic and hormonal traits regarding the growth and productivity-associated characteristics of escarole plants by the inoculation with AMF and the mulching treatment while maintaining statistical variability [37]. The explained variability, which is the sum of that from the principal component (PC) 1 and PC2, was over 50%. Importantly, the loading PCA revealed that the growth parameters (shoot FW, root FW, leaf number, and leaf area) were closely associated with each other and with important hormonal (JA, total GAs, and total CKs) factors (Figure 8B). Furthermore, NUE and PUE were coupled with all growth variables studied. In contrast, the ionic (leaf and root NO_3_^−^, leaf and root P^5+^, and leaf Cu^2+^, Zn^2+^, and Mg^2+^) and the ethylene precursor ACC were inversely associated with the productivity traits of escarole plants (Figure 8B).

## 3. Discussion

With the rapid increase in the world’s population, there is a corresponding rise in food demand which is followed by concerns about the stability of the global environment [38]. Some strategies such as the use of AMF or mulching technology have been used to improve plant growth and performance and crop productivity [11,39,40,41]. The combined use of AMF and ecological mulching could be a sustainable strategy to increase plant production in horticultural crops, especially in the Mediterranean basin which is one of the most important horticultural areas in Europe. Previously, we have described growth developmental changes in mulched escarole plants grown under water-limiting conditions [20]. We found that ecological hydromulching improved growth and productivity under water stress, and this was related to the capacity of the plant to regulate water relations and CO_2_ assimilation through fine-tuning stomata opening due to the antagonistic interaction of CKs and ABA [20]. In the present study, escarole plants were assayed under the presence or absence of the AMF *Rhizophagus irregularis*, and with different mulching treatments. We have found that the inoculation with AMF increased the growth of escarole plants in all mulching treatments, thus indicating a synergistic effect of AMF and (hydro)mulching. Importantly, the MS treatment had an additional positive effect, especially in shoot and root FW and leaf area (Figure 2C–E). Several studies confirm that mycorrhized roots can explore more soil volume due to their extramatrical hyphae, which facilitate the absorption and translocation of several nutrients [42,43]. Indeed, increments in phosphorous content in plant tissues have been reported due to the enhanced uptake by the hyphae [12,44]. In our study, phosphorous concentrations in both the shoot and root combined in an opposite cluster to that of the growth parameters within the PCA (Figure 8). Despite the lower phosphorous concentrations in mycorrhized plants by the use of Hoagland-P^low^ (Table 1 and Table 2), *R. irregularis* improved phosphorous uptake in escarole plants, thus maintaining and/or increasing plant growth (Figure 2). It has been documented that under nutrient limitations plants increase their radicular structure to facilitate ionic absorption and avoid nutrient imbalance [45], while other authors also reported that the root system morphological development can be affected by soil nutrient status and also by AMF [46]. Furthermore, the use of organic mulches is also related to a higher area of the root zone [47]. This could explain the increase in most macro- and micronutrients in the roots of AMF plants, especially in those hydromulched with MS (Table 2). Importantly, the growth improvement provoked by AMF inoculation (Table 3) was associated with phosphorous and nitrogen use efficiencies in the loading PCA (Figure 8B). The study of Zhu et al. [48] informed that wheat plants grown under elevated CO_2_ levels presented significant increases in their nitrogen use efficiency when these plants were inoculated with AMF. Furthermore, Liu et al. [49] demonstrated that AMF symbiosis had a positive effect on the nitrogen use efficiency of soybean grown under water stress.

In addition to providing nutritional and structural benefits to plants, AMF also awards other benefits including production/accumulation of secondary metabolites, increased resistance against biotic and abiotic stresses, and enhanced photosynthesis rates [50,51,52]. According to Begum et al. [11], increased photosynthetic activities are directly related to the improved growth frequency of AMF inoculation. Indeed, in our study, AMF inoculation and mulching synergistically increased the gas exchange parameters of escarole plants, especially at the end of the experimental period (Figure 3). As stated above, the highest gas exchange-related parameters in PE-mulched plants may be due to the physical properties of this material. The impermeability of PE blocked substrate evaporation, and, thus, these plants had to increase stomatal opening to facilitate plant water (and gas) exchange, resulting in a rise in photosynthetic activity. Recently, Song et al. [40] have shown in wheat mulched plants that the use of plastic films strongly increases the net photosynthetic rate. Niu et al. [53] also described the positive effect of plastic mulch on photosynthetic rates and dark respiration rates, due mainly to the higher stomatal density and aperture area of maize leaves.

Importantly, the growth and productivity promotion associated with the interaction of AMF inoculation and mulching could be explained by a direct effect on the hormonal balance of the plant. Indeed, we found the lowest endogenous ABA levels in plants mulched with PE treatment and inoculated with AMF (Figure 7B), which presented the highest stomatal conductance (Figure 3B). Lettuce plants colonized by *R. intraradices* have shown lower ABA levels than non-AMF plants, thus maintaining stomatal opening under saline conditions [54]. In contrast, the study of Khalloufi et al. [19] informed about the increase in endogenous ABA concentrations in tomato plants inoculated with *R. irregularis*, both under control and saline conditions. Taking this into account, it might be suggested that the effect of AMF on ABA content could vary depending on the host plant and/or the experimental conditions.

Furthermore, in our study, the growth-related hormones GAs and CKs were also affected, thus enhancing plant development (Appendix A and Figure 8B). A specific mulching response was found regarding GA leaf content (Figure 5 and Appendix A), especially in MS mulching treatment. In this regard, the growth-related parameters clustered with GA3, GA4, and total GAs in the loading PCA (Figure 8B). Recently, the study of Romero-Muñoz et al. [20] has stated that bioactive GA levels increased in escarole plants mulched with MS. In addition, early reports in tobacco plants have revealed that AMF is also related to the production of some types of GAs, such as GA1 and its deactivation product GA8 [22]. Likewise, the study of Khalloufi et al. [19] found that the levels of GA1 and GA3 increased in leaves of mycorrhized tomato plants, both under control and saline conditions. Therefore, the interaction between GAs and AMF seems to be especially important to increase plant growth. CKs are classically closely related to plant growth responses [55] and were also especially affected by the interaction of AMF inoculation and mulching. Indeed, the concentration of one of the most active CKs in plants, tZ, significantly increased in mycorrhized plants (Figure 6A and Appendix A), especially in the AMF plants mulched with MS. Interestingly, these plants showed the highest shoot, root, and total FW alone or in combination with AMF inoculation (Figure 2B–D). Importantly, CKs also clustered with the growth-related parameters in the loading PCA (Figure 8B). Early reports observed a strong correlation between enhanced growth, improved photosynthesis, and the increase in CK levels in AMF plants [22,56]. The study of Cosme et al. [24] suggested that both shoot- and root-specific alterations of endogenous CK levels had key roles in the relation between CK homeostasis and growth in tobacco plants inoculated with AMF. Similarly, Miransari et al. [29] reported that increased phosphate uptake in mycorrhized plants could be explained by the augmentation of root CK content and CK flux to the shoots, thus enhancing plant growth.

Several authors have addressed the possible role of auxins in plant growth [19,23]. However, in the present study, we found that endogenous levels of IAA in escarole leaves were not affected by either AMF inoculation or the mulching treatment. Importantly, the endogenous levels of SA and JA increased with the use of mulching, but, notably, after AMF inoculation, especially in the MS treatment (Figure 7D,E). We found that JA clustered with the growth parameters (Figure 8B), but the correlation between SA and the growth-related parameters was unclear. This suggests a role for JA in the control of escarole plant growth. In this regard, some previous reports have shown the positive effect of JA on growth in AMF plants [29,52]. Nevertheless, both SA and JA are related to biotic stress responses, indicating that the fungal hyphae colonization may provoke plant defense responses, which are compensated for by the additional benefits on plant growth [16].

## 4. Materials and Methods

### 4.1. Biological Material and Experimental Design

The AM fungus Rhizophagus irregularis (Błaszk., Wubet, Renker and Buscot) C. Walker and A. Schüßler as [‘irregulare’] was obtained from the collection of the Experimental Field Station of Zaidín, Granada, Spain (EEZ 6). The AM fungal inoculum consisted of a mixture of rhizospheric soil from pot cultures (*Sorghum* spp.) that contained spores, hyphae, and mycorrhizal root fragments, with a potential infectivity of ∼35 infective propagules per gram of inoculum [57]. For the mycorrhizal (M) treatment, 10 g of the substrate was mixed with the AMF inoculum substrate (coconut fiber/vermiculite, 1/2, *w*/*w*). The inoculum was placed adjacent to the root of the seedling. For the non-mycorrhized (NM) treatment, the substrate was prepared in the same final ratio (1:2) without AM fungal inoculum. Mycorrhizal (M) and non-mycorrhizal (NM) escarole plants (*Cichorium endivia* L.) three weeks old plants cv. Bekele were transplanted to 2.5-L containers. Two mulching treatments were installed on top of the substrate: the hydromulching treatment, based on a substrate used for mushroom cultivation (MS), and low-density black polyethylene (PE). The non-covered treatment (BS) was used as a control. The escarole plants were irrigated regularly with Hoagland nutrient solution [58]. A modified Hoagland solution (i.e., 70% P-impoverished solution, referred to as Hoagland-P^low^ throughout the text) was used to irrigate the mycorrhized plants. The irrigation was performed by self-compensating drippers (2 L h^–1^) and a fresh nutrient solution was applied to avoid nutrient imbalance. The amount of water used to irrigate the plants was adjusted every week, depending on the demand of the plants. Plants were grown in a climate-controlled growth chamber under a 16 h daylight period. The air temperature ranged from 22 to 25 °C during the day. Relative humidity was maintained at 70 ± 5% and the light intensity was approximately 200 μmol m^−2^ s^−1^. Escarole plants were harvested 55 days after transplantation. The experiment was designed as a factorial combination of AM fungal inoculation treatment (NM and M) and the use of different mulching treatments (BS, PE, and MS), which entailed six combined treatments in total.

### 4.2. Plant Growth-Related Determinations

Plant growth-related parameters were recorded at the end of the experiment in 4 plants per mulching treatment in M and NM plants. Plants were washed with distilled water and separated into shoots and roots. The parameters recorded were shoot fresh weight (FW), root FW, leaf number, and leaf area (LA). Total fresh weight (TFW) was calculated as the sum of root and shoot FW. LA was quantified using an LI-COR leaf area meter (Model LI-3100C; LI-COR, Lincoln, NE, USA).

### 4.3. AM Fungal Root Colonization

Roots were sampled and analyzed via ink staining [59]. The roots were cut into small pieces and placed in 50-ML plastic tubes (Sarstedt, Nümbrecht, Germany). Twenty-five mL of KOH 10% was added to the roots before incubation at 70 °C in a water bath for 45 min. The KOH was then removed, and roots were washed with HCl 1%. The staining step consisted of adding 25 mL of ink 2% (Parker Pen Company, Nantes, France) in HCl 1%. The tubes were then placed at 70 °C in a water bath for 30 min. The roots were rinsed and stored in deionized water before analysis. The total colonization rate was quantified using the gridline intersect method [60]. Positive counts for AM colonization included the presence of vesicles or arbuscules or typical mycelium within the roots.

### 4.4. Gas Exchange Measurements

Gas exchange was monitored in fully expanded leaves at the plant’s vegetative stage. Net CO_2_ fixation rate (Amax, µmol CO_2_ m^−2^ s^−1^), stomatal conductance to water vapor (gs, mmol H_2_O m^−2^ s^−1^), and transpiration rate (E, mmol H_2_O m^−2^ s^−1^) were measured in a steady state under conditions of saturating light (800 µmol m^−2^ s^−1^) and 400 ppm CO_2_ with an LI-6400 instrument (LI-COR, Lincoln, NE, USA). The intrinsic water use efficiency (WUEi) of leaf gas exchange was calculated from the gas exchange data as Amax/E.

### 4.5. Chlorophyll Content

Chlorophylls were extracted from 1 g of frozen escarole leaves (−80 °C) with 25 mL of acetone solvent. Samples were homogenized and centrifuged at 5000× *g* for 6 min at 4 °C. Subsequently, the optical density of the supernatant was measured spectrophotometrically at wavelengths of 663 and 645 nm. The contents of chlorophyll a and b were calculated according to the Nagata and Yamashita equations [61]:Chlorophyll a (mg·100 mL^−1^) = 0.999 × A663 − 0.0989 × A645
Chlorophyll b (mg·100 mL^−1^) = −0.328 × A663 + 1.77 × A645

### 4.6. Maximum Potential Quantum Efficiency of PSII

On the leaf used for gas exchange, the ratio between the variable fluorescence from a dark-adapted leaf (Fv) and the maximal fluorescence from a dark-adapted leaf (Fm), which is called the maximum potential quantum efficiency of PSII (Fv/Fm), was determined with a portable modulated fluorometer, model OS-30P (Opti-Science, Hudson, NY, USA). This ratio is one of the most widely used in research employing the fluorescence technique and is considered a good indicator of the in vivo functionality of the photosynthetic apparatus in plants [62]. A special leaf clip holder was allocated to each leaf to maintain dark conditions for at least 30 min before reading.

### 4.7. Plant Mineral Content

Full-sized leaves and root fragments of each plant were freeze-dried for 72 h at −55 °C (Christ Alpha 1-2 LDplus, Osterode am Harz, Germany). Anions were extracted with bidistilled water and were subsequently measured with an ion chromatograph (METROHM 861 Advanced Compact IC; METROHM 838 Advanced Sampler); the column used was a METROHM Metrosep CARB1 150/4.0 mm. Cations were extracted by acid digestion, using an ETHOSONE microwave digestion system (Milestone Inc., Shelton, CT, USA) and analyzed by inductively coupled plasma optical emission (ICP-OES, Varian Vista MPX, Palo Alto, CA, USA). Nutrient use efficiency (NUE) was calculated as the ratio between total plant dry weight (g) and total nutrient content in the plant (g) according to Gerloff and Gabelman [63] in Baligar et al. [64], as follows:$$NUE=\frac{Plant\;weight\;\left(g\right)}{Nutrient\;content\;\left(g\right)\;}\;$$

### 4.8. Hormone Extraction and Analysis

Cytokinins (trans-zeatin, tZ, zeatin riboside, ZR, and isopentenyl adenine, iP), gibberellins (GA1, GA3, and GA4), indole-3-acetic acid (IAA), abscisic acid (ABA), salicylic acid (SA), jasmonic acid (JA), and the ethylene precursor 1-aminocyclopropane-1-carboxylic acid (ACC) were analyzed according to Albacete et al. [65] and Großkinsky et al. [66] with some modifications. Briefly, a total of 100 mg of plant material was grounded in liquid nitrogen and dropped in 0.5 mL of cold (−20 °C) extraction mixture of methanol/water (80/20, *v*/*v*). Then, 10 µL of an internal standard mix, composed of deuterated hormones ([2H5]tZ, [2H5]tZR, [2H6]iP, [2H2]GA1, [2H2]GA3, [2H2]GA4, [2H5]IAA, [2H6]ABA, [2H4]SA, [2H6]JA, [2H4]ACC, Olchemim Ltd., Olomouc, Czech Republic) at a concentration of 1 µg mL-1 each, were added to the extraction homogenate. Solids were separated by centrifugation (20,000× *g*, 15 min, 4 °C) and re-extracted for 30 min at 4 °C in an additional 0.5 mL of the same extraction solution. Pooled supernatants were passed through Sep-Pak Plus C18 cartridges (SepPak Plus, Waters, Milford, MA, USA) to remove interfering lipids and part of the plant pigments and evaporated at 40 °C under vacuum to near dryness. The residue was dissolved in 0.2 mL methanol/water (20/80, *v*/*v*) solution using an ultrasonic bath. The dissolved samples were filtered through 13 mm diameter Millex filters with 0.22 µm pore size nylon membrane (Millipore, Bedford, MA, USA). A total of 10 µL of filtered extract were injected in a U-HPLC-MS system consisting of an Accela Series U-HPLC (ThermoFisher Scientific, Waltham, MA, USA) coupled to an Exactive mass spectrometer (ThermoFisher Scientific, Waltham, MA, USA) using a heated electrospray ionization (HESI) interface. Mass spectra were obtained using the Xcalibur software version 2.2 (ThermoFisher Scientific, Waltham, MA, USA). For the quantification of the plant hormones, calibration curves were constructed for each analyzed component (1, 10, 50, and 100 µg L^−1^) and corrected for 10 µg L^−1^ deuterated internal standards. Recovery percentages ranged between 92 and 95%.

### 4.9. Statistical Analyses

The data were tested first for homogeneity of variance and normality of distribution. The significance of the treatment effects was determined by analysis of variance (ANOVA). The significance (*p* ≤ 0.05) of the differences between mean values was tested by Tukey’s honestly significant difference (HSD). Principal component analyses (PCA) were also performed. The Varimax rotation method was used for the loading PCA, while the score PCA was graphically plotted as a biplot score. All statistical analyses were conducted using the IBM SPSS software (Version 25.0, IBM SPSS Corp., Chicago, IL, USA).

## 5. Conclusions

In this study, we have tested the growth effects of a new mulching formulation called hydromulching in combination with the inoculation of the arbuscular mycorrhizal fungus *Rhizophagus irregularis* in a commercial escarole variety grown under controlled conditions. The mulching application and the AMF inoculation separately, but especially interactively, enhanced the growth of escarole plants. The growth improvement observed has been explained by the regulation of nutrient use efficiency and the hormonal balance of the plant. In this regard, mycorrhized plants presented higher NUE and PUE, especially those covered with hydromulch. Furthermore, inoculated plants mulched with MS increased the most active GAs, especially GA3, which clustered with all growth-related parameters studied. Likewise, the active CKs, tZ and iP, and JA concentrations increased with AMF inoculation or hydromulching application, but particularly by their interaction, which was associated with plant growth promotion. Hence, this work is an important step toward the use of sustainable strategies to improve horticultural crop production. This is especially relevant in the actual climate change context since there exists an urgent need for sustaining food security, particularly in the Mediterranean basin which is one of the most important horticultural areas in Europe.

## Figures and Tables

**Figure 1 plants-11-02795-f001:**
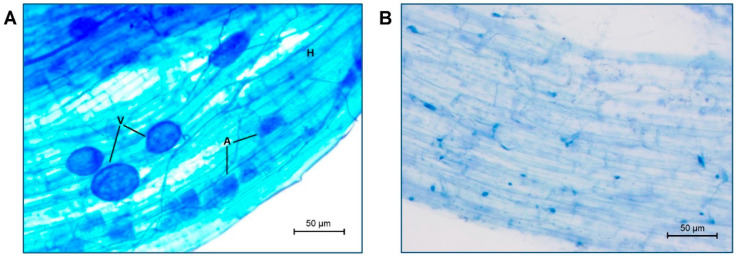
Microphotographs of (**A**) mycorrhized and (**B**) non-mycorrhized escarole roots at the end of the experiment. Abbreviations used: vesicles (V), arbuscules (A), and hyphae (H).

**Figure 2 plants-11-02795-f002:**
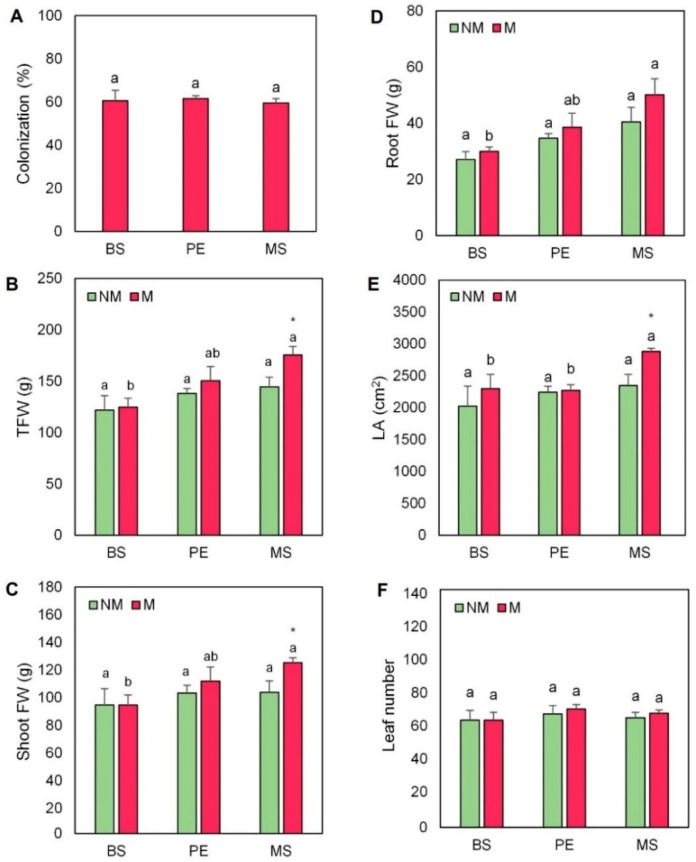
(**A**) Percentage of colonization, (**B**) total fresh weight (FW), (**C**) shoot FW, (**D**) root FW, **(E**) leaf area (LA), and (**F**) leaf number of escarole plants of the commercial variety “Bekele” cultivated under non-mycorrhizal (NM) and mycorrhizal (M) conditions and covered or not with mulching treatments. Bars show the means of five plants ± standard error. Different letters show significant differences among mulching treatments while the asterisk (*) indicates significant differences due to AMF inoculation within each mulching treatment according to Tukey’s and *t*-tests (*p* ≤ 0.05). Abbreviations used: bare soil (BS), polyethylene mulch (PE), and mushroom substrate-based hydromulch (MS).

**Figure 3 plants-11-02795-f003:**
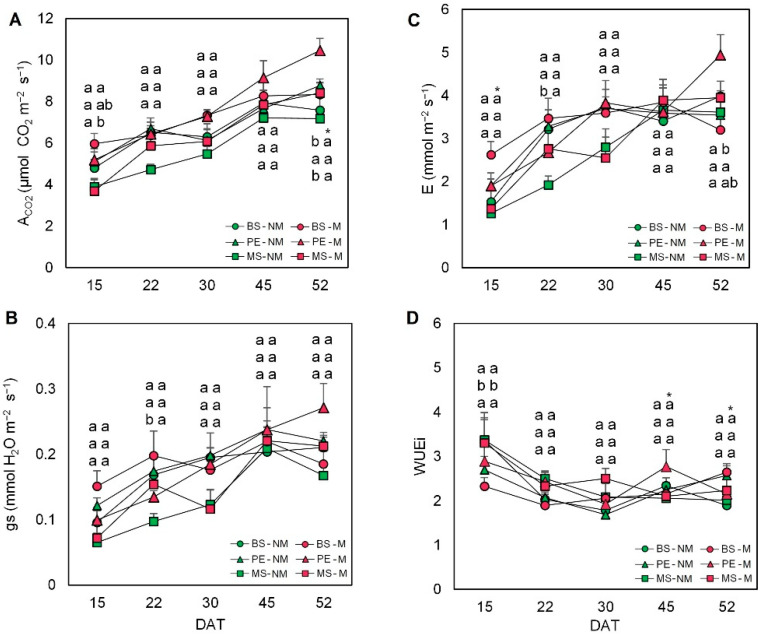
(**A**) Evolution of photosynthetic rate (A_CO_2__), (**B**) stomatal conductance (gs), (**C**) transpiration rate (E), and (**D**) intrinsic water use efficiency (WUEi) in escarole plants of the commercial variety “Bekele” cultivated under non-mycorrhizal (NM) and mycorrhizal (M) conditions and covered or not with mulching treatments. Bars show the means of five plants ± standard error. Different letters show significant differences among mulching treatments while the asterisk (*) indicates significant differences due to AMF inoculation within each mulching treatment according to Tukey’s and *t*-tests (*p* ≤ 0.05). Abbreviations used: days after transplant (DAT), bare soil (BS), polyethylene mulch (PE), and mushroom substrate-based hydromulch (MS).

**Figure 4 plants-11-02795-f004:**
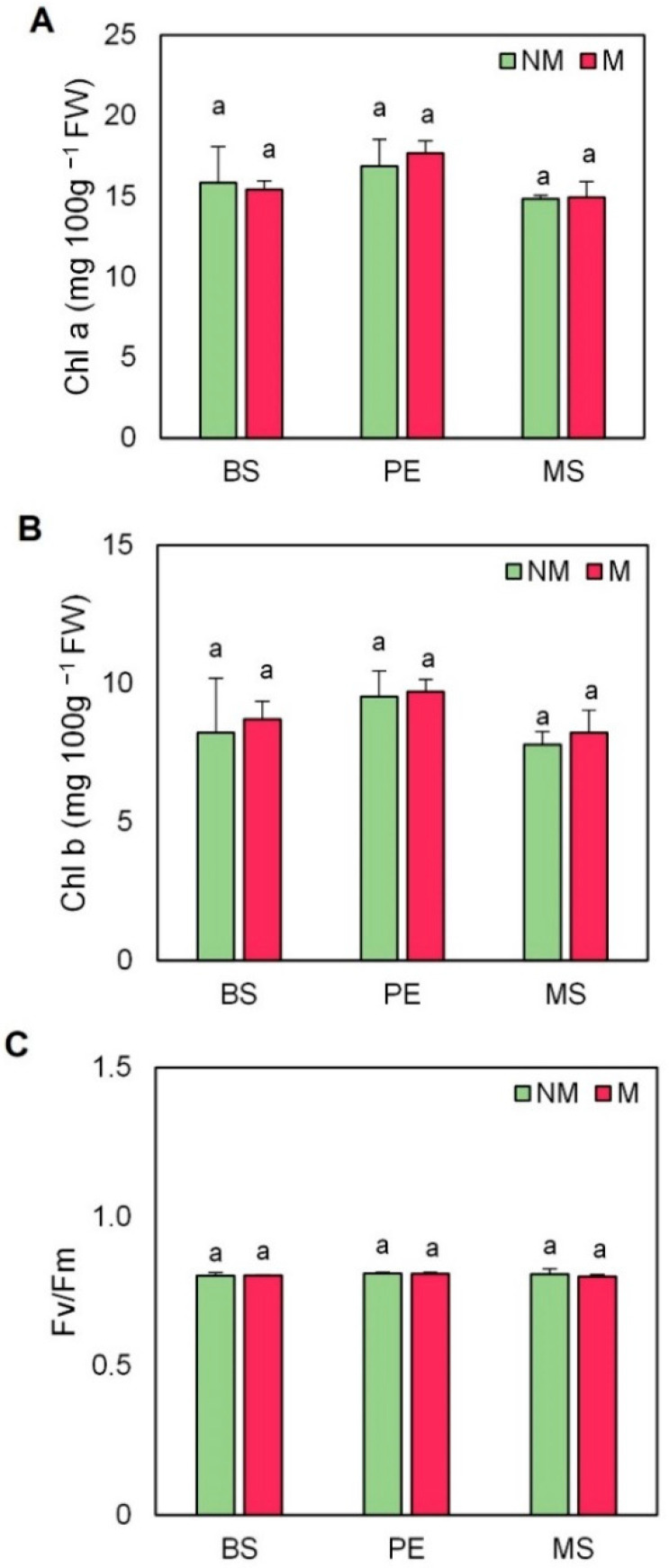
(**A**) Chlorophyll a, (**B**) chlorophyll b, and (**C**) chlorophyll fluorescence (Fv/Fm) in the leaves of escarole plants of the commercial variety “Bekele” cultivated under non-mycorrhizal (NM) and mycorrhizal (M) conditions and covered or not with mulching treatments. Bars show the means of five plants ± standard error. Different letters show significant differences among mulching treatments while the asterisk (*) indicates significant differences due to AMF inoculation within each mulching treatment according to Tukey’s and *t*-tests (*p* ≤ 0.05). Abbreviations used: bare soil (BS), polyethylene mulch (PE), and mushroom substrate-based hydromulch (MS).

**Figure 5 plants-11-02795-f005:**
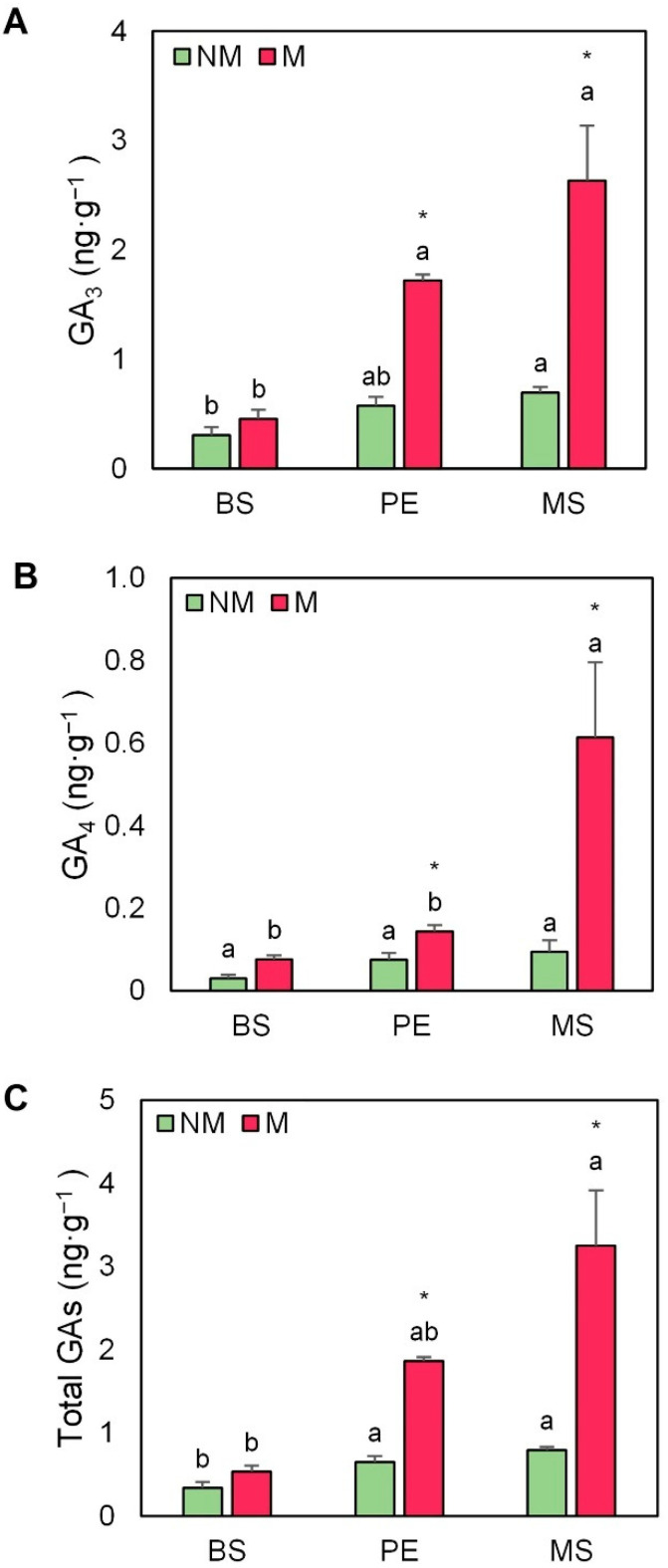
(**A**) Gibberellin A3 (GA3), (**B**) gibberellin A4 (GA4), and (**C**) total gibberellin (GAs) concentrations in leaves of escarole plants of the commercial variety “Bekele” cultivated under non-mycorrhizal (NM) and mycorrhizal (M) conditions and covered or not with mulching treatments. Bars show the means of five plants ± standard error. Different letters show significant differences among mulching treatments while the asterisk (*) indicates significant differences due to AMF inoculation within each mulching treatment according to Tukey’s and *t*-tests (*p* ≤ 0.05). Abbreviations used: bare soil (BS), polyethylene mulch (PE), and mushroom substrate-based hydromulch (MS).

**Figure 6 plants-11-02795-f006:**
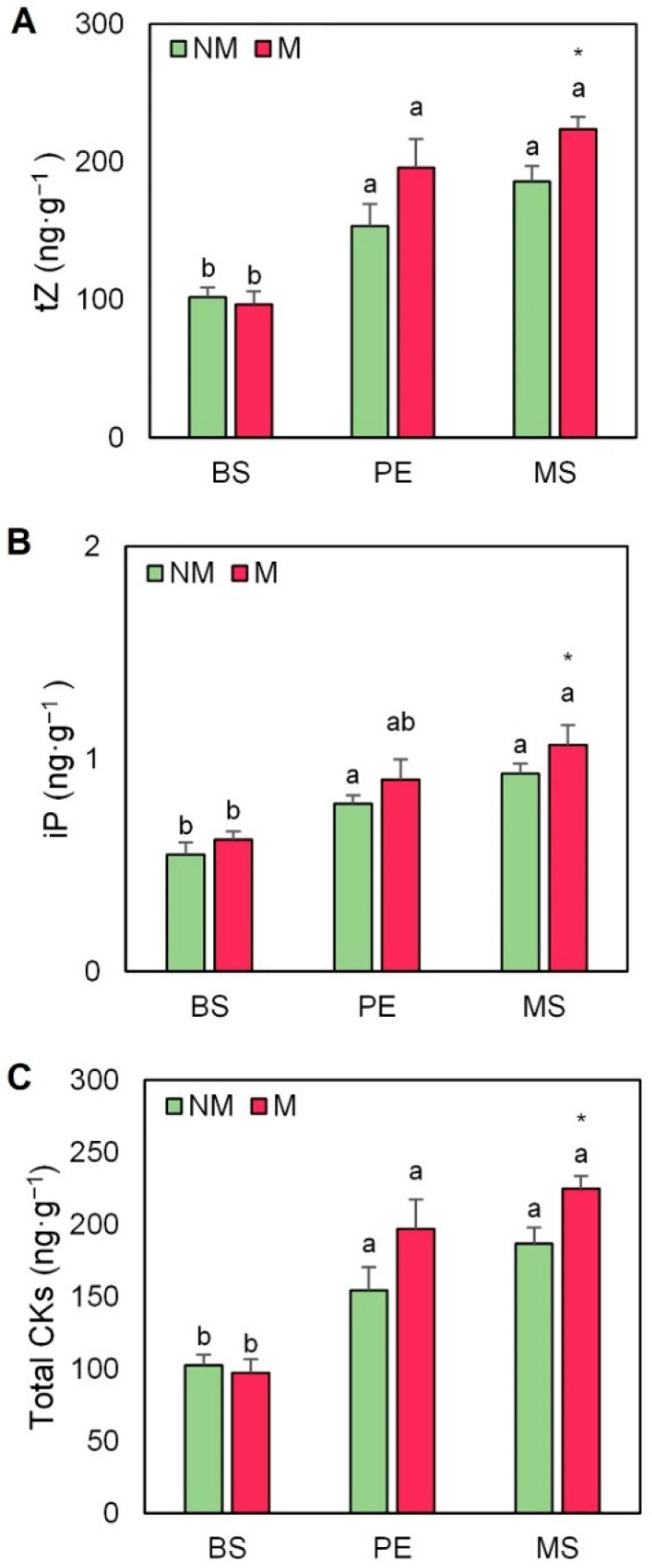
(**A**) Trans-zeatin (tZ), (**B**) isopentenyladenine (iP), and (**C**) total cytokinin (CKs) concentrations in leaves of escarole plants of the commercial variety “Bekele” cultivated under non-mycorrhizal (NM) and mycorrhizal (M) conditions and covered or not with mulching treatments. Bars show the means of five plants ± standard error. Different letters show significant differences among mulching treatments while the asterisk (*) indicates significant differences due to AMF inoculation within each mulching treatment according to Tukey’s and *t*-tests (*p* ≤ 0.05). Abbreviations used: bare soil (BS), polyethylene mulch (PE), and mushroom substrate-based hydromulch (MS).

**Figure 7 plants-11-02795-f007:**
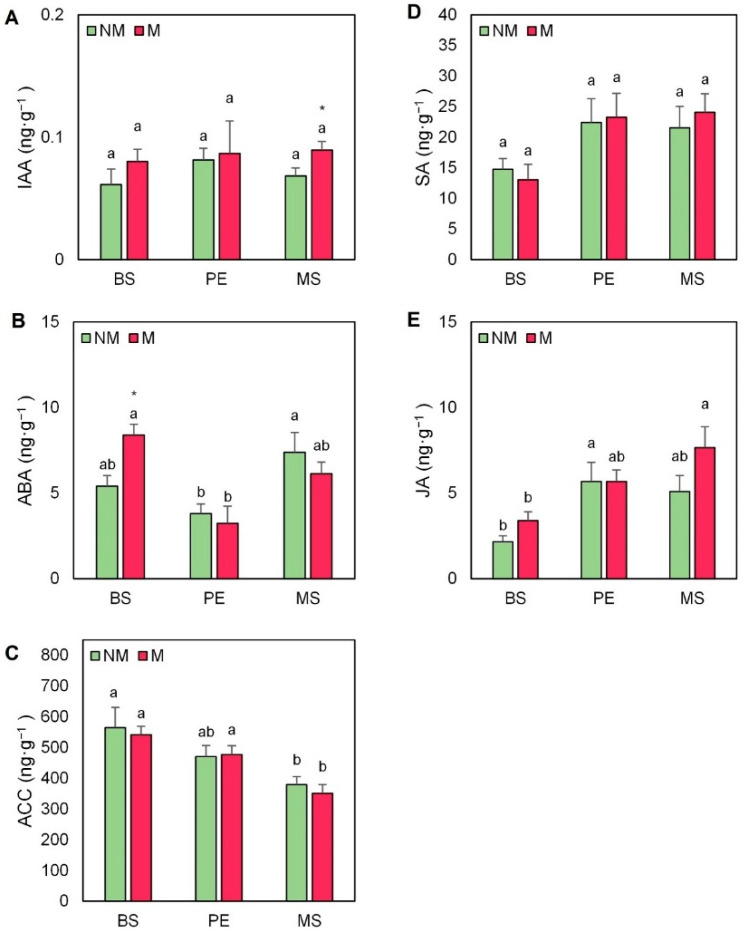
(**A**) Indole acetic acid (IAA), (**B**) abscisic acid (ABA), (**C**) 1-aminocyclopropane-1-carboxylic acid (ACC), (**D**) salicylic acid (SA), and (**E**) jasmonic acid (JA) concentrations in leaves of escarole plants of the commercial variety “Bekele” cultivated under non-mycorrhizal (NM) and mycorrhizal (M) conditions and covered or not with mulching treatments. Bars show the means of five plants ± standard error. Different letters show significant differences among mulching treatments while the asterisk (*) indicates significant differences due to AMF inoculation within each mulching treatment according to Tukey’s and *t*-tests (*p* ≤ 0.05). Abbreviations used: bare soil (BS), polyethylene mulch (PE), and mushroom substrate-based hydromulch (MS).

**Figure 8 plants-11-02795-f008:**
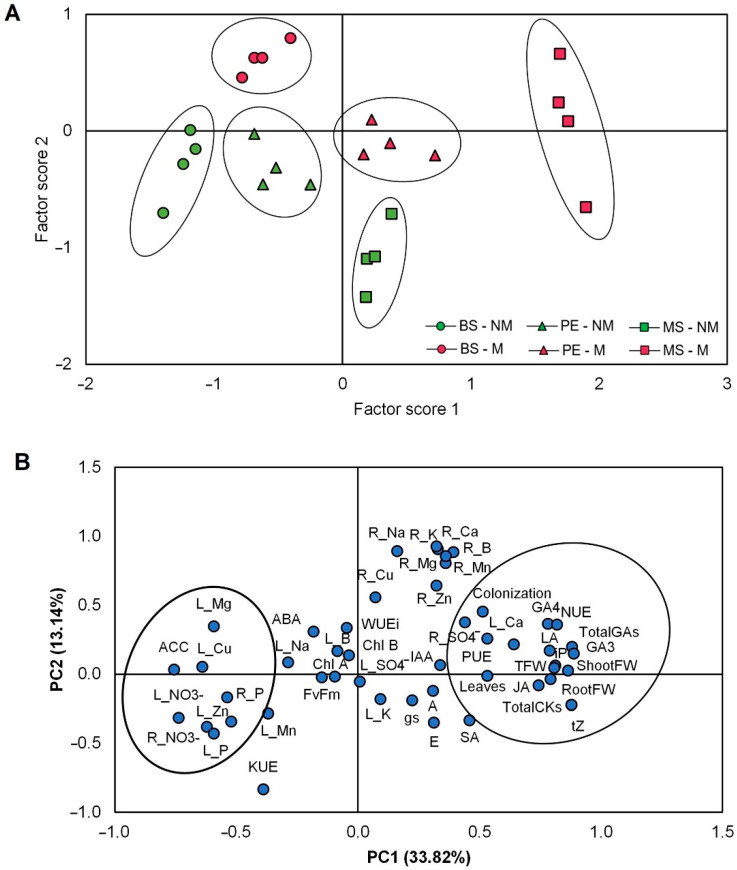
(**A**) Biplot representing the score values and (**B**) two axes of a principal component (PC1 and PC2) analysis showing the loadings of various growth-related, ionic, and hormonal variables (denoted by abbreviations) of the escarole commercial variety “Bekele” non-mulched or subjected to different mulching treatments and cultivated under non-mycorrhizal (NM) and mycorrhizal (M) conditions. Circles enclose those variables/scores that cluster together in loading PCA and score PCA. Abbreviations used: boron (B^3+^), calcium (Ca^2+^), copper (Cu^2+^), potassium (K^+^), magnesium (Mg^2+^), manganese (Mn^2+^), sodium (Na^+^), phosphorus (P^5+^), sulfate (SO_4_^2−^), nitrate (NO_3_^−^), nitrogen use efficiency (NUE), phosphorous use efficiency (PUE), potassium use efficiency (KUE), root fresh weigh (RootFW), shoot fresh weight (ShootFW), leaf area (LA), total fresh weight (TFW), chlorophyll a (Chl a), chlorophyll b (Chl b), net CO_2_ fixation rate (A), stomatal conductance (gs), transpiration rate (E), intrinsic water use efficiency (WUEi), abscisic acid (ABA), 1-aminocyclopropane-1-carboxylic acid (ACC), indole acetic acid (IAA), salicylic acid (SA), jasmonic acid (JA), gibberellin A1 (GA1), gibberellin A4 (GA4), total gibberellins (Gas), trans-zeatin (tZ), isopentenyladenine (iP), and total cytokinins (CKs).

**Table 1 plants-11-02795-t001:** Mineral nutrient concentrations in leaves of escarole plants (cv. Bekele) cultivated under non-mycorrhizal (NM) and mycorrhizal (M) conditions and covered or not with mulching treatments.

**AMF**	**Mulch**	**NO_3_^−^** **(mg g^−1^ DW)**	**P^5+^** **(mg g^−1^ DW)**	**K^+^** **(mg g^−1^ DW)**	**Mg^2+^** **(mg g^−1^ DW)**	**Ca^2+^** **(mg g^−1^ DW)**	**SO_4_^−^^2^** **(mg g^−1^ DW)**
NM	BS	22.83 ± 4.18 a *	5.65 ± 0.44 a *	51.51 ± 1.91 b	6.06 ± 0.64 a *	6.44 ± 0.85 a	7.81 ± 1.83 a
	PE	14.22 ± 3.15 a	5.57 ± 0.40 a *	54.50 ± 1.22 ab	5.35 ± 0.80 a	7.35 ± 0.54 a	7.56 ± 1.03 a
	MS	12.84 ± 2.10 a*	5.07 ± 0.29 a *	58.44 ± 1.87 a	3.80 ± 0.14 a	8.33 ± 0.22 a	7.64 ± 0.95 a
M	BS	7.60 ± 0.45 a	3.14 ± 0.32 a	54.20 ± 2.68 a	5.89 ± 0.62 a	6.72 ± 0.93 a	9.52 ± 1.06 a
	PE	7.19 ± 0.92 a	2.84 ± 0.34 a	54.54 ± 2.33 a	4.96 ± 0.34 ab	6.86 ± 0.33 a	12.67 ± 1.67 a *
	MS	2.06 ± 0.05 b	2.71 ± 0.27 a	53.99 ± 2.90 a	3.84 ± 0.29 b	9.18 ± 0.81 a	7.61 ± 1.32 a
**AMF**	**Mulch**	**Cu^2+^** **(mg kg^−1^ DW)**	**Mn^2+^** **(mg kg^−1^ DW)**	**Zn^2+^** **(mg kg^−1^ DW)**	**B^3+^** **(mg kg^−1^ DW)**	**Na^+^** **(mg g^−1^ DW)**	
NM	BS	6.43 ± 0.96 a	44.75 ± 50.92 a	95.34 ± 10.03 a	37.55 ± 3.03 a	6.33 ± 0.97 a	
	PE	5.47 ± 0.38 a	34.99 ± 32.19 a	90.74 ± 9.91 a	38.42 ± 4.79 a *	6.67 ± 1.11 a *	
	MS	4.65 ± 0.22 a	42.06 ± 47.53 a	78.72 ± 5.49 a	34.93 ± 5.49 a	5.23 ± 0.34 a	
M	BS	7.20 ± 0.72 a *	41.12 ± 27.92 a	100.49 ± 10.44 a	27.48 ± 4.37 a	4.81 ± S0.83 a	
	PE	5.90 ± 0.29 ab	32.85 ± 25.06 a	87.25 ± 13.54 a	32.02 ± 4.44 a	3.46 ± 0.35 a	
	MS	4.25 ± 0.78 b	31.49 ± 35.15 a	64.15 ± 9.41 a	36.74 ± 3.64 a	5.12 ± 0.92 a	

Different letters within a column show significant differences among mulching treatments while the asterisk (*) indicates significant differences due to AMF inoculation within each mulching treatment according to Tukey’s and *t*-tests (*p* ≤ 0.05). Abbreviations used: boron (B^3+^), calcium (Ca^2+^), copper (Cu^2+^), potassium (K^+^), magnesium (Mg^2+^), manganese (Mn^2+^), nitrate (NO_3_^-^), sodium (Na^+^), phosphorous (P^5+^), sulphate (SO_4_^2^^−^), zinc (Zn^2+^), bare soil (BS), polyethylene mulch (PE), and mushroom substrate-based hydromulch (MS).

**Table 2 plants-11-02795-t002:** Mineral nutrient concentrations in roots of escarole plants (cv. Bekele) cultivated under non-mycorrhizal (NM) and mycorrhizal (M) conditions and covered or not with mulching treatments.

**AMF**	**Mulch**	**NO_3_^−^** **(mg g^−1^ DW)**	**P^5+^** **(mg g^−1^ DW)**	**K^+^** **(mg g^−1^ DW)**	**Mg^2+^** **(mg g^−1^ DW)**	**Ca^2+^** **(mg g^−1^ DW)**	**SO_4_^−^^2^** **(mg g^−1^ DW)**
NM	BS	5.53 ± 0.56 a	7.26 ± 0.91 a *	41.49 ± 5.10 a	4.76 ± 0.73 a	6.19 ± 0.71 a	14.91 ± 1.34 b
	PE	5.97 ± 0.77 a	8.44 ± 0.32 a *	51.27 ± 1.12 a	5.28 ± 0.24 a	6.66 ± 0.32 a	12.32 ± 0.69 b
	MS	4.08 ± 1.08 a *	3.92 ± 0.38 b *	39.58 ± 3.49 a	4.16 ± 0.48 a	5.99 ± 0.68 a	22.24 ± 0.71 a
M	BS	2.61 ± 0.08 a	1.93 ± 0.49 a	70.39 ± 1.52 a	7.45 ± 1.29 a	8.54 ± 1.63 a	18.64 ± 0.80 ab *
	PE	2.66 ± 0.23 a	1.45 ± 0.10 a	50.76 ± 3.87 a	5.52 ± 0.22 a	7.40 ± 0.44 a	14.11 ± 1.25 b *
	MS	1.22 ± 0.17 b	2.08 ± 0.36 a	87.47 ± 1.49 a *	8.36 ± 1.10 a *	10.33 ± 1.37 a *	24.18 ± 2.56 a
**AMF**	**Mulch**	**Cu^2+^** **(mg kg^−1^ DW)**	**Mn^2+^** **(mg kg^−1^ DW)**	**Zn^2+^** **(mg kg^−1^ DW)**	**B^3+^** **(mg kg^−1^ DW)**	**Na^+^** **(mg g^−1^ DW)**	
NM	BS	20.56 ± 1.03 ab	81.23 ± 5.94 a	30.03 ± 2.89 a	13.76 ± 2.26 a	8.29 ± 0.81 ab	
	PE	22.72 ± 8.83 a	68.12 ± 4.95 ab	38.22 ± 0.70 a	18.10 ± 0.66 a	9.12 ± 0.33 a	
	MS	12.65 ± 0.85 b	59.77 ± 2.93 a	29.37 ± 3.24 a	12.35 ± 1.31 a	6.53 ± 0.62 b	
M	BS	27.37 ± 6.25 a	114.85 ± 8.25 a	34.85 ± 2.67 a	21.65 ± 2.77 a	11.19 ± 2.51 a	
	PE	35.42 ± 7.85 a	63.75 ± 5.20 a	39.15 ± 3.77 a	20.57 ± 0.57 a	10.22 ± 0.72 a	
	MS	22.75 ± 2.77 a *	187.35 ± 10.11 a *	45.85 ± 1.35 b	25.47 ± 1.81 a *	11.07 ± 1.68 a *	

Different letters within a column show significant differences among mulching treatments while the asterisk (*) indicates significant differences due to AMF inoculation within each mulching treatment according to Tukey’s and *t*-tests (*p* ≤ 0.05). Abbreviations used: boron (B^3+^), calcium (Ca^2+^), copper (Cu^2+^), potassium (K^+^), magnesium (Mg^2+^), manganese (Mn^2+^), nitrate (NO_3_^−^), sodium (Na^+^), phosphorous (P^5+^), sulphate (SO_4_^2^^−^), zinc (Zn^2+^), bare soil (BS), polyethylene mulch (PE), and mushroom substrate-based hydromulch (MS).

**Table 3 plants-11-02795-t003:** Nutrient use efficiency of escarole plants (cv. Bekele) cultivated under non-mycorrhizal (NM) and mycorrhizal (M) conditions and covered or not with mulching treatments.

AMF	Mulch	NUE	PUE	KUE
NM	BS	25.03 ± 2.58 b	802.94 ± 93.98 b	108.74 ± 6.90 a
	PE	39.02 ± 6.32 ab	717.66 ± 29.99 b	94.66 ± 1.88 a
	MS	59.28 ± 7.97 a	1127.48 ± 75.91 a	102.22 ± 2.56 a *
M	BS	90.88 ± 3.73 b *	1985.08 ± 82.96 a *	83.02 ± 8.21 a
	PE	91.08 ± 15.97 b *	2346.22 ± 105.89 a *	95.19 ± 2.74 a
	MS	335.93 ± 26.96 a *	1290.88 ± 280.68 a *	74.01 ± 9.04 a

Different letters within a column show significant differences among mulching treatments while the asterisk (*) indicates significant differences due to AMF inoculation within each mulching treatment according to Tukey’s and *t*-tests (*p* ≤ 0.05). Abbreviations used: nitrogen use efficiency (NUE), phosphorous use efficiency (PUE), potassium use efficiency (KUE), bare soil (BS), polyethylene mulch (PE), and mushroom substrate-based hydromulch (MS).

## Data Availability

All data will be available under reasonable request to the corresponding author.

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
