# Peer review of "The Interaction between Hydromulching and Arbuscular Mycorrhiza Improves Escarole Growth and Productivity by Regulating Nutrient Uptake and Hormonal Balance"

_plants, 2022, doi:10.3390/plants11202795_

Round 1

Reviewer 1 Report

The work of Romero-Muñoz M. with co-authors is devoted to the study of a complex analysis of synergistic effects of inoculation by AMP fungi and hydromulching on growth and physiological and biochemical indicators of escarol plants. Much work has been done to study sparse growth, the components of photosynthesis and respiration, the state of the food elements and the hormonal status of escarole plants.   The relevance of the work is clear, as the application of environmentally sound approaches to increase the yield of vegetable crops is an important strategic objective.  In the process of reading the manuscript there were some comments, the correction of which will greatly improve the work and make it more clear for perception.

The opening lines of the Introduction and Discussion sections are very similar.  In the Introduction section you can explain why this culture is chosen - escarol plants.   Emphasize that the important culture of the Mediterranean region is being studied

Lines 182-183 is already «Discussion» and not «Results».  Under the tables the subtypical signatures and decipherments should be distinguished from the whole text .

The article states the synergetic effect and we want to focus on this in the Discussion. In the Discussion everything is right and a lot. But please highlight the «red» points of this effect.

Reviewer 2 Report

The ms plants-1964105 with the title of The interaction between ecological hydromulching and arbuscular mycorrhizal fungi improves the growth and productivity of escarole (Chichorium endivia L.) by regulating nutrient use efficiency and hormonal balance investigates an interesting topic, but the authors have to improve it before it can be accepted in such high-quality journal.

Please follow the point of my comment by point:

Title: make it shorter.

Title: Please use the English name of the Latin name of the plant

Title: use arbuscular mycorrhiza instead of arbuscular mycorrhizal fungi

L37-42 please cite this ref: https://doi.org/10.1007/978-3-030-64323-2_17

In the results section, it would be great if the authors can present some pictures of the Mycorrhizal colonization with roots.

Table 1. Why do you thing that Mn is higher than other micro elements in the leaves of escarole?

In Table 1, I am wondering why N and P in leaves are higher in non-mycorrhizal (NM) than mycorrhizal (M)? Something is wrong? Accordng to my info, increasing the colonization with AMF will enhance the plants to uptake the P, and N. Please see  https://doi.org/10.1016/j.chemosphere.2013.02.004

The statistical analysis should be improved in terms of presentation because it is hard to be followed by readers. Authors in some figures do not have to present the capital letters or the SE can be enough to show the significance.

The order of presenting the statistics on the figures is not clear, therefore the authors should reorder the columns within each figure to make it easier for the readers. Please try different types of presenting your data and then select the best one.

L428 gr should be corrected to g

Why authors did not conduct such an experiment in the open field? It would be great

L528 < should be

L532 SPSS Inc is wrong, this company was sold to another company. Please replace the old name with the new name of the company or the owner.

In the conclusion, please add the most important finding in terms of some values.

Regards, Reviewer

Round 2

Reviewer 2 Report

Thank you for revising your ms.